# Geographical Gradients of Genetic Diversity and Differentiation among the Southernmost Marginal Populations of *Abies sachalinensis* Revealed by EST-SSR Polymorphism

**Keiko Kitamura [1], Kentaro Uchiyama [2], Saneyoshi Ueno [2], Wataru Ishizuka [3], Ikutaro Tsuyama [1] and Susumu Goto [4,***

[1] Hokkaido Research Center, Forestry and Forest Products Research Institute, 7 Hitsujigaoka, Toyohira, Sapporo, Hokkaido 062-8516, Japan; kitamq@ffpri.affrc.go.jp (K.K.); itsuyama@affrc.go.jp (T.I.)

[2] Department of Forest Molecular Genetics and Biotechnology, Forestry and Forest Products Research Institute, 1 Matsunosato, Tsukuba, Ibaraki 305-8687, Japan; kruchiyama@affrc.go.jp (U.K.); saueno@ffprc.affrc.go.jp (U.S.)

[3] Forestry Research Institute, Hokkaido Research Organization, Koushunai, Bibai, Hokkaido 079-0166, Japan; wataru.ishi@gmail.com

[4] Education and Research Center, The University of Tokyo Forests, Graduate School of Agricultural and Life Sciences, The University of Tokyo, 1-1-1 Yayoi, Bunkyo-ku, Tokyo 113-8657, Japan

* Correspondence: gotos@uf.a.u-tokyo.ac.jp

**Abstract:** *Research Highlights*: We detected the longitudinal gradients of genetic diversity parameters, such as the number of alleles, effective number of alleles, heterozygosity, and inbreeding coefficient, and found that these might be attributable to climatic conditions, such as temperature and snow depth. *Background and Objectives*: Genetic diversity among local populations of a plant species at its distributional margin has long been of interest in ecological genetics. Populations at the distribution center grow well in favorable conditions, but those at the range margins are exposed to unfavorable environments, and the environmental conditions at establishment sites might reflect the genetic diversity of local populations. This is known as the central-marginal hypothesis in which marginal populations show lower genetic variation and higher differentiation than in central populations. In addition, genetic variation in a local population is influenced by phylogenetic constraints and the population history of selection under environmental constraints. In this study, we investigated this hypothesis in relation to *Abies sachalinensis*, a major conifer species in Hokkaido. *Materials and Methods:* A total of 1189 trees from 25 natural populations were analyzed using 19 EST-SSR loci. *Results:* The eastern populations, namely, those in the species distribution center, showed greater genetic diversity than did the western peripheral populations. Another important finding is that the southwestern marginal populations were genetically differentiated from the other populations. *Conclusions:* These differences might be due to genetic drift in the small and isolated populations at the range margin. Therefore, our results indicated that the central-marginal hypothesis held true for the southernmost *A. sachalinensis* populations in Hokkaido.

**Keywords:** central-marginal hypothesis; cline; Pinaceae; trailing edge population; Sakhalin fir; sub-boreal forest

## 1. Introduction

The spatial distribution of genetic variation provides essential information for conservation programs and management of forest tree species [1]. The geographic distribution center provides favorable biotic and abiotic environments for species persistence. On the contrary, peripheral

populations are exposed to unfavorable environmental conditions and the populations at the range margin are smaller and more isolated from each other than those at the range core [2–4]. The central-marginal hypothesis states that marginal populations show lower genetic variation and higher differentiation than do central populations (see review [5,6]). The above evidence was obtained from a study of Scots pine [7] and silver fir (*Abies alba*), which showed a decline in gene diversity among the western margin populations [8].

Long-lived plant species are sessile organisms and their survival is attributed to their adaptation to the local environment. Such adaptation processes inevitably involve genetic changes in local populations [9]. However, genetic variation in a local population is influenced by phylogenetic constraints and the population history of selection under environmental constraints. Genetic markers are, therefore, useful for inferring the genetic structure of local populations. They are also useful for inferring adaptability to similar environments according to similarities in genetic characteristics, as geographic genetic variations reflect the results of selection and adaptation to the local environment [10].

Sub-boreal conifer species play an important role in the sustainability of its forest ecosystems across the northern hemisphere. *Abies sachalinensis* is a major sub-boreal conifer in East Asia. The geographical distribution of *A. sachalinensis* ranges from Sakhalin in the north, to the southern Kuril Islands in the east, and Hokkaido, the northernmost island of the Japanese Archipelago, in the south [11,12]. The species often occurs across a broad altitudinal range; namely, from sea level to 1500 m a.s.l. It is one of the major conifers of the montane forest in Hokkaido and often forms mixed forests with other conifers, such as *Picea jezoensis*, *P. glehnii*, and broadleaf species, such as *Quercus mongolica* var. *crispula*, *Betula ermanii*, *Fagus crenata*, and *Tilia japonica*. At its southern and western distribution, *A. sachalinensis* is scarce, whereas it is abundant in the northern and eastern regions [13,14]. However, the distribution center of *A. sachalinensis*, including genetic core populations, has not yet been determined to date.

The species has been known to show both regional geographic and altitudinal variations in morphological traits [15–17]. Kurahashi and Hamaya [15] earlier noticed a wide variety of altitudinal differences in growth traits. This phenomenon was later revealed to be the result of adaptation to the local environment of the establishment sites [18]. Ishizuka et al. [19] confirmed that the altitudinal gradient of the autumn phenology related to cold tolerance was genetically controlled. Further, Goto et al. [20] found natural selection at the QTL (quantitative trait loci) of phenological traits across altitudinal differences. Hatakeyama [16] reported that regional differences in morphological traits were closely related to the snow-related climatic conditions based on a common garden experiment. Further, Eiga [17] reported regional genetic clines in both the altitudinal differences and longitudinal range differences in cold tolerance in Hokkaido, which are attributable to the level of snow acclimation at the establishment sites. A longitudinal gradient was also observed in allozyme variations [21]. Moreover, Suyama et al. [22] suggested a single lineage based on the cpDNA sequence of *A. sachalinensis* among Hokkaido populations. However, due to the limited numbers of populations and loci, the regional genetic relationships among *A. sachalinensis* populations in Hokkaido have not yet been resolved.

In this study, we investigated regional geographic variations in *A. sachalinensis* in Hokkaido by EST-SSR polymorphism with a special emphasis on the southern range marginal population. We sought to answer two questions; is the central-marginal hypothesis applicable to *A. sachalinensis*? And does the environmental gradient affect genetic clines as indicated by Nagasaka et al. [21]?

## 2. Materials and Methods

We chose 25 natural populations of *A. sachalinensis* from across the island of Hokkaido and a total of 1189 mature individuals as sample trees (Figure 1, Table 1). Total DNA was extracted from 100 mg of fresh needle leaves by the DNeasy Plant Mini Kit (QIAGEN K. K., Tokyo, Japan).

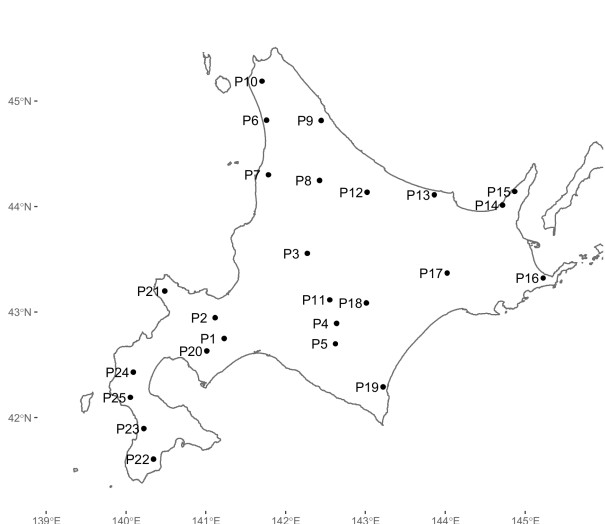

**Figure 1.** Locations of the 25 natural populations of *Abies sachalinensis* used in this study.

**Table 1.** Study sites of *Abies sachalinensis* populations in Hokkaido, northern Japan.

| Population ID | Longitude East (°) | Latitude North (°) | Alt [1] (m) | N [2] | Nttl_a [3] | Na [4] | Nef [5] | $H_O$ | $H_E$ | $G_{IS}$ |
|---|---|---|---|---|---|---|---|---|---|---|
| P1 | 141.2365 | 42.7691 | 800 | 44 | 68 | 3.579 | 1.815 | 0.385 | 0.383 | −0.006 |
| P2 | 141.1201 | 42.9663 | 580 | 48 | 66 | 3.474 | 1.975 | 0.379 | 0.410 | 0.074 |
| P3 | 142.3221 | 43.5685 | 300 | 50 | 75 | 3.947 | 1.871 | 0.383 | 0.401 | 0.044 |
| P4 | 142.6851 | 42.9003 | 930 | 47 | 75 | 3.947 | 1.928 | 0.408 | 0.417 | 0.021 |
| P5 | 142.6656 | 42.7072 | 700 | 48 | 73 | 3.842 | 1.962 | 0.419 | 0.418 | −0.003 |
| P6 | 141.8080 | 44.8361 | 5 | 46 | 70 | 3.684 | 1.862 | 0.356 | 0.382 | 0.067 |
| P7 | 141.8255 | 44.3187 | 90 | 48 | 71 | 3.737 | 1.978 | 0.394 | 0.409 | 0.038 |
| P8 | 142.4986 | 44.2581 | 285 | 48 | 72 | 3.789 | 1.964 | 0.419 | 0.409 | −0.025 |
| P9 | 142.5356 | 44.8255 | 260 | 48 | 67 | 3.526 | 1.908 | 0.392 | 0.386 | −0.016 |
| P10 | 141.7521 | 45.2064 | 10 | 48 | 66 | 3.474 | 1.916 | 0.389 | 0.395 | 0.014 |
| P11 | 142.6033 | 43.1250 | 360 | 43 | 68 | 3.579 | 1.930 | 0.387 | 0.396 | 0.024 |
| P12 | 143.1218 | 44.1365 | 250 | 48 | 76 | 4.000 | 2.000 | 0.399 | 0.422 | 0.054 |
| P13 | 144.0025 | 44.0930 | 200 | 48 | 73 | 3.842 | 1.903 | 0.395 | 0.400 | 0.014 |
| P14 | 144.8912 | 43.9676 | 73 | 48 | 70 | 3.684 | 1.965 | 0.376 | 0.411 | 0.085 |
| P15 | 145.0595 | 44.0907 | 100 | 47 | 77 | 4.053 | 1.916 | 0.355 | 0.399 | 0.110 |
| P16 | 145.3731 | 43.2588 | 20 | 48 | 70 | 3.684 | 1.888 | 0.382 | 0.407 | 0.062 |
| P17 | 144.1321 | 43.3482 | 950 | 48 | 77 | 4.053 | 1.975 | 0.377 | 0.416 | 0.093 |
| P18 | 143.0752 | 43.0874 | 750 | 48 | 75 | 3.947 | 1.904 | 0.392 | 0.409 | 0.041 |
| P19 | 143.2620 | 42.2881 | 73 | 48 | 75 | 3.947 | 2.022 | 0.380 | 0.424 | 0.102 |
| P20 | 141.0124 | 42.6518 | 450 | 48 | 69 | 3.632 | 1.887 | 0.393 | 0.402 | 0.023 |
| P21 | 140.4678 | 43.2183 | 200 | 48 | 62 | 3.263 | 1.934 | 0.425 | 0.385 | −0.105 |
| P22 | 140.3397 | 41.6239 | 90 | 48 | 47 | 2.474 | 1.768 | 0.312 | 0.313 | 0.001 |
| P23 | 140.2135 | 41.9132 | 470 | 48 | 62 | 3.263 | 1.803 | 0.379 | 0.374 | −0.014 |
| P24 | 140.0727 | 42.4468 | 620 | 48 | 67 | 3.526 | 1.897 | 0.389 | 0.393 | 0.010 |
| P25 | 140.0379 | 42.2073 | 100 | 48 | 66 | 3.474 | 1.939 | 0.378 | 0.384 | 0.017 |
| **Overall** | | | | 1189 | | | | 0.386 | 0.398 | 0.030 |

[1] Altitude, [2] number of individuals analyzed, [3] total number of alleles, [4] number of alleles, [5] effective number of alleles.

We analyzed the previously reported 11 EST-SSR loci: Aat01, Aat02, Aat04, Aat05, Aat06, Aat08, Aat09, Aat10, Aat11, Aat13, Aat15 [23]. In addition, we developed 8 new EST-SSR markers from *A. sachalinensis* transcriptome data. All the transcript sequences (158,542) from TodoFirGene [24] were used as input for the CMIB (CD-HIT-EST, MISA, ipcress and BlastCLUST) pipeline [25] to obtain PCR primers for amplifying unique microsatellite sequences with the number of repeat units ≥ 6, 5, 4, 3, and 3 for di-, tri-, tetra-, penta-, and hexa-simple sequence repeats (SSRs), respectively. For each primer pair, genomic DNA from one individual was used to check PCR amplification. PCR reactions were carried out following the standard protocol included in the QIAGEN Multiplex PCR Kit (QIAGEN, Hilden, Germany). For the primer pairs that exhibited clear microsatellite peaks at the expected fragment length, the extracted DNA of 32 individuals of *A. sachalinensis* representative of the species' range was used to evaluate EST-SSR polymorphism. Among them, 8 polymorphic markers, Egm1005,

Egm14860, Egm16822, Egm26233, Egm4191, Egm4389, Egm5979, and Egm55338, were used for the following analysis. Genotyping data has been deposited in the TreeGenes Database under accession number TGDR252.

PCR was carried out with a Type-it Microsatellite PCR Kit (QIAGEN K. K., Tokyo, Japan). Fragment analyses were performed by ABI 3130-xl Genetic Analyzer, 600 LIZ size standard, and GENESCAN for Windows (Thermo Fischer Scientific, Tokyo, Japan).

We used GenoDive ver. 2.0b17 [26] to calculate the following genetic parameters; observed ($H_O$), expected heterozygosity ($H_E$), total heterozygosity ($H_T$), inbreeding coefficient ($G_{IS}$), genetic differentiation ($G'_{ST}$), number of alleles (Na), effective number of alleles (Nef), and Nei's genetic distance (D). Principal component analysis (PCA) was conducted to clarify the genetic relationship among populations using the same software. Four genetic diversity parameters; number of alleles, effective number of alleles, $H_E$, and $G_{IS}$, were then spatially interpolated by kriging [27] using R [28] and laid out on the contour map. A neighbornet phylogenetic tree based on Nei's genetic distance was drawn using SplitsTree4 ver. 4.14.6 [29].

The climatic conditions for each population were estimated by the Mesh Climate Data 2000 [30]. We calculated the following environmental factors; WI, warmth index; CI, cold index; TMC, mean minimum temperature of the coldest month; PRS, precipitation in summer (May to September); PRW, precipitation in winter (October to April); MSD, maximum snow depth; WinSR, solar radiation in winter (October to April); SprSR, solar radiation in spring (May); SumSR, solar radiation in summer (June to August); and AutSR, solar radiation in autumn (September). Classification of four seasons for solar radiation was determined based on the regression coefficient ($r > 0.7$) between months (Table S1).

## 3. Results

The 19 EST-SSR loci used in this study were polymorphic, with the number of alleles ranging from 2 to 13 (Table S2). The genetic diversity ($H_E$) of the 25 populations ranged from 0.313 to 0.424, and the overall $H_E$ was 0.398. The $G'_{ST}$ ranged from −0.105 to 0.110, and the overall value was 0.016 (Table S2). Each genetic diversity parameter, along with the 95% confidence interval for the 25 populations, is shown in Figure S1. The effective number of alleles and $H_E$ did not differ significantly among the populations. The number of alleles of P22 was significantly lower than those of P3, P4, P5, P12, P15, P17, P18, and P19, and the $G_{IS}$ of P21was significantly lower than that of P15 (Figure S1). The geographic patterns of genetic diversity were represented by contour diagrams (Figure 2). All four parameters showed longitudinal gradients; that is, the eastern populations showed higher values for the genetic diversity parameters (Figure 3). In addition, the four genetic parameters for each locus are shown in Figures S2–S5. Most of the loci showed eastward increases in genetic parameters, but several of them, for example, Aat09, Aat10, and Egm1005, showed opposite results for the effective number of alleles and $H_E$ (Figures S2 and S4).

Pairwise differentiation matrices between populations by $G'_{ST}$ (Table S3) revealed significant differentiation between the isolated populations (P19, P22) and southwestern populations (P20–21, P23–25) and the rest of the populations.

Principle component analysis by covariance matrix revealed that the plots of the geographically peripheral populations (P19 to P25) were in outlying positions on the first and second axis plane (Figure 4). The plots of the southern populations, in particular, showed smaller Co1 and larger Co2 scores (grouped within the dotted line in Figure 4). P19 and P21 showed the highest Co1 and the lowest Co2 scores, respectively. The other populations did not show any geographical clustering and were located at the center of the axis plane.

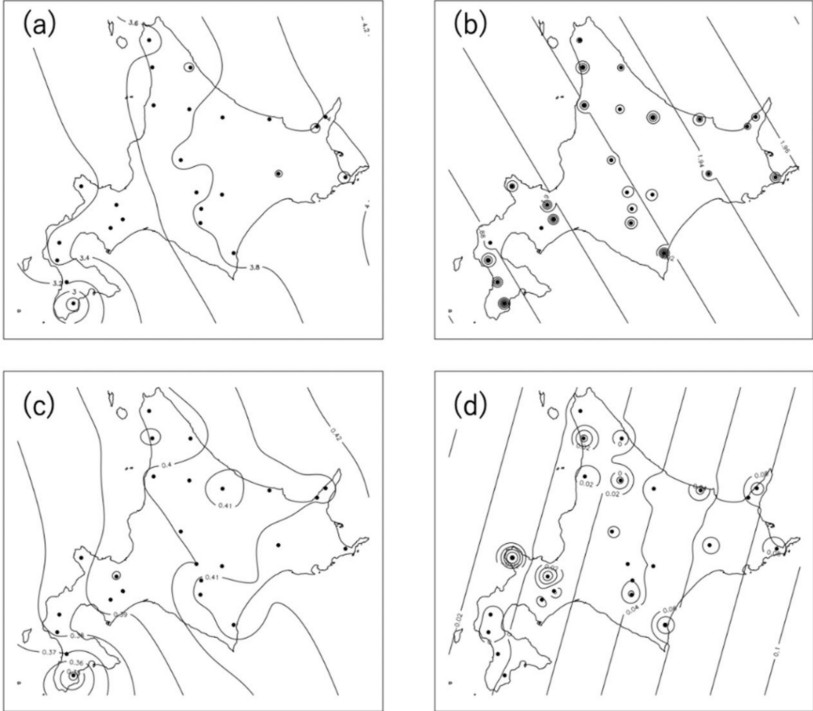

**Figure 2.** Spatial contour maps of four genetic parameters, the number of alleles (**a**), effective number of alleles (**b**), $H_E$ (**c**), and $G_{IS}$ (**d**), measured for the 25 natural populations. Dots indicate study sites (cf., Table 1, Figure 1).

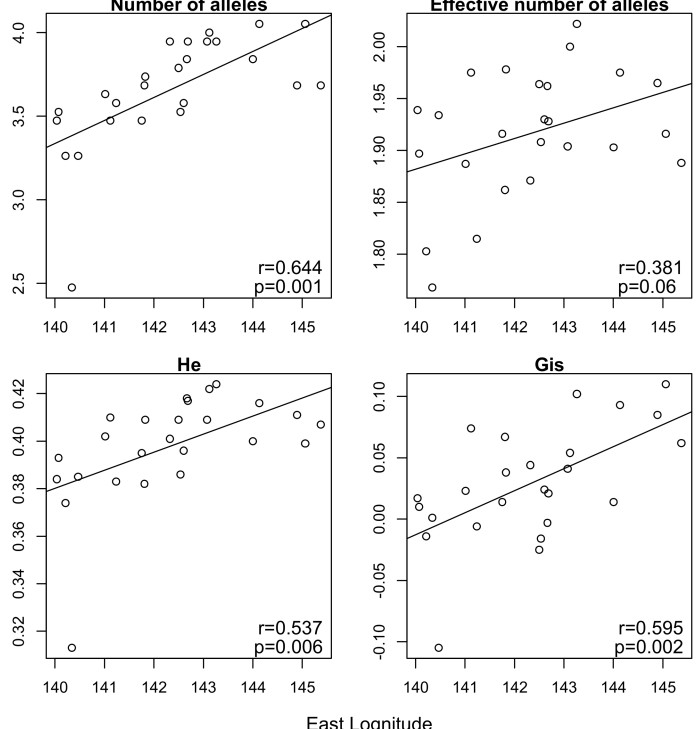

**Figure 3.** Relationships among genetic parameters and longitude east. Lines indicate the least squares.

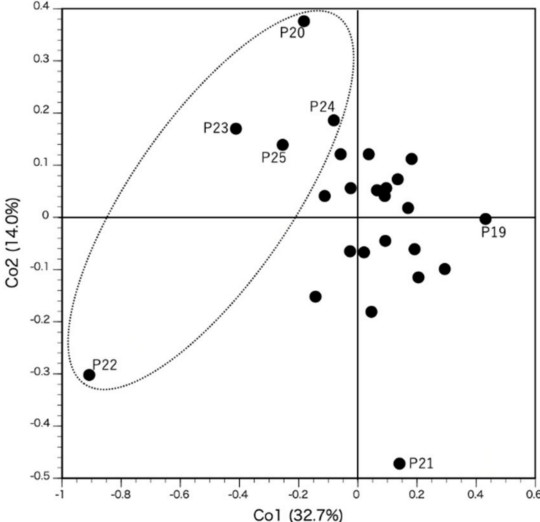

**Figure 4.** Principle component analysis based on the covariance matrix of the 25 populations. The first (Co1) and second (Co2) axes' % of variances are in parentheses. Southern range populations are grouped by the dotted line.

The phylogenetic relationships of the 25 populations are shown by neighbornet tree based on Nei's genetic distances (Figure 5). The tree shows that the plots of the geographically peripheral populations (P19 to P25) were in outlying positions. Moreover, the southern populations were placed on the same branch (grouped within the dotted line in Figure 5). The southernmost population (P22) was located at the furthest position in the cluster made up of the southern populations. Other geographically peripheral populations, namely, P19 and P25, were clustered to individual branches and were differentiated from the central populations.

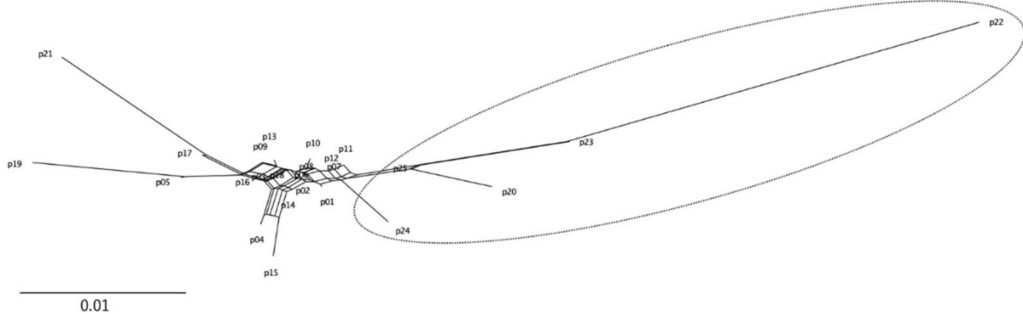

**Figure 5.** Neighbornet tree among the 25 populations based on Nei's genetic distances. Southern range populations are grouped by the dotted line.

An environmental gradient along longitude was found among the climatic conditions for *A. sachalinensis* populations. The results from the correlation test (Table S4) and principal component analysis (Figure 6) between longitude and climatic conditions indicated negative correlations between longitude and winter precipitation (PRW), maximum snow depth (MSD), mean minimum temperature of the coldest month (TMC), and solar radiation in autumn (AurSR), as well as a positive correlation between longitude and solar radiation in winter (WinSR). Although PRW showed a significant *p*-value alone after Bonferroni correction (Table S4), the climatic conditions for the 25 populations studied along longitude can be assumed; thus, the eastern populations are characterized by less snow, colder winters, and more solar radiation in winter. On the contrary, the western populations have more snow, warmer winters, and less solar radiation in winter. We did not detect any relationship between solar radiation and precipitation in summer.

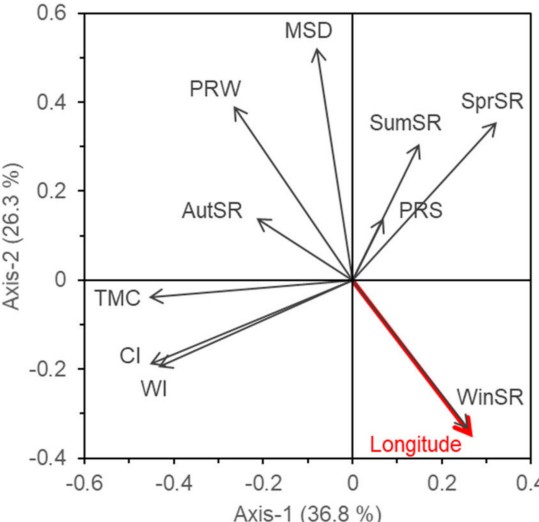

**Figure 6.** Principal component analysis using longitude and selected environmental factors for 25 populations of *A. sachalinensis*.

In addition, we detected significant relationships between Na and climatic environmental factors (Table S5). We observed that the number of alleles at 10 loci showed significant relationships with climatic factors, with seven of these loci showing significant relationships with more than two factors.

## 4. Discussion

Central populations are characterized by high genetic diversity due to a large effective population size. In contrast, peripheral populations are often characterized by a small effective population size. Our present results regarding genetic diversity parameters; namely, Na, Nef, and $H_E$, were greater in the northeastern than in the southwestern populations (Figures 2a–c and 3). This indicated that the distribution center of *A. sachalinenesis* in eastern Hokkaido has greater genetic diversity than do the southwestern peripheral populations. The species distribution model indicated that the suitable habitat for *A. sachalinenesis* was from central to eastern Hokkaido, and the marginal habitat was on the southwestern Oshima Peninsula [31]. Thus, the northern and eastern parts of Hokkaido can be regarded as the distribution center of this species, as corroborated by the cumulative volume of *A. sachalinensis* in previous studies [13,14]. The island of Hokkaido includes the southernmost distribution range of *A. sachalinensis* on its southwestern peninsula. Similar results were obtained for European silver fir, *A. alba*, which has a long pollen flow distance similar to *A. sachalinensis* and shows a decline in gene diversity in its western marginal populations [8]. Thus, the central-marginal hypothesis is supported in *A. sachalinensis* in this study.

On the one hand, the inbreeding coefficient, $G_{IS}$, also showed the same gradient (Figures 2d and 3). The reason for this tendency has not yet been clarified, but the same tendency has been reported in previous studies of marginal populations of the anemogamous species, *Fagus crenata* [4]; that is, the smaller the population, the lower the inbreeding coefficient.

Previous studies of *A. sachalinensis* revealed east-to-west longitudinal genetic clines in morphological traits, resistance against disease, tolerance against environmental conditions by provenance tests [16,17] and allozymes polymorphisms among natural populations [21]. The present study also revealed the same gradient in genetic parameters determined by EST-SSRs. Nagasaka et al. [21] detected significant east–west directional variation patterns in $H_E$, effective number of alleles, and number of alleles based on 4 allozyme loci, and they regarded this variation to be due to temperature and precipitation. A longitudinal gradient in gene diversity can be attributable to the climatic factors at the establishment site [32]. A longitudinal gradient in climatic factors in Hokkaido

might involve temperature and snow depth [17,33], with eastern Hokkaido being colder but having less accumulation of snow than southwestern Hokkaido. Okada et al. [34] revealed that the number of winter bud scales of *A. sachalinensis* differed between the eastern and western populations, and they inferred that this indicated a response to drought hardiness in the winter season. Eiga [17] indicated a longitudinal gradient in freezing resistance in seedlings. These morphological traits might result from the natural selection and adaptation to the local climate, which was one of the major causes of the longitudinal gradient in genetic diversity.

Our results showed that some environmental factors for the 25 populations showed a longitudinal gradient (Table S4, Figure 6). The relationships between the number of alleles and climatic factors were revealed to be significant for 10 EST-SSR loci (Table S5). Among them, the most highly correlated climatic factor was TMC, with the second most being PRW, while PRS and SumSR did not show any significance. This result indicated that the climate conditions in winter were more correlated to genetic traits than were those in summer. Specifically, colder temperatures, less snow, and more solar radiation in the winter and spring were indicative of a greater number of alleles within a population. Several loci were correlated with more than one climatic factor, such as Aat02, Egm26233, and Egm4389 with WI, TMC, and PRW. These climatic factors also showed longitudinal clines (Table S5). Thus, the specific relationships between loci and climate factors could be reflected in the regional differences in gene diversity observed in *A. sachalinensis* in this study.

Previous studies have revealed relationships between climatic conditions and gene diversity in natural populations of forest tree species. For example, associations between AFLP loci and temperature were observed in *Fagus sylvatica* [35] and *Betula pendula* [36]. Grivet et al. [37] found two SNPs that were correlated with temperature in *Pinus pinaster* and *P. halepensis*. Annotated EST-SSR loci in *Eucalyptus gomphocephala* showed clines in allele frequencies for climatic factors, such as solar radiation, potential evaporation, summer precipitation, and aridity [38]. Former studies of allozyme variation also revealed adaptive differentiation in conifer species (e.g., [39]). Not only allele frequencies but also the heterozygosity of an individual is often associated with fitness among a changing environment (e.g., [40]). Correlations have been reported between growth rate and heterozygosity in *Populus tremuloides* [41] and survivorship and heterozygosity in *Picea jezoensis* [42]. It has been observed that the heterozygosity of a population influences its productivity, fitness, and stability [40,43,44]. Indeed, genetic diversity was affected by many other factors, such as demographic history and genetic drift, and we have to wait for association analyses, such as the outlier test, to reveal the natural selection of certain alleles against the environment for *A. sachalinensis*.

Another interesting issue regarding the longitudinal gradient was the observed counteraction among individual loci (Figures S2–S4). Most of the loci showed higher genetic diversity in the eastern than in the western populations, but several loci showed an opposite trend. This might offer evidence of opposite selection or adaptation forces to the environmental gradient between different loci. Currently, we are not able to clarify this discrepancy, but it is likely that these loci were affected by the selective sweep of adaptive genes [45,46] as the loci used in this study were EST-SSR, whose primers were developed among the sequences in close proximity to expressed gene sequences.

STRUCTURE analysis [47] by F-model with 70,000 burn-in and 30,000 MCMC detected differences in the admixture coefficient between the southwestern and the other populations (Figure S6) but did not indicate any regionally specific ancestral clusters among the 25 populations. This might be attributable to the long pollen flow distance of anemogamous species such as *A. sachalinensis*. A long distance gene flow may result in gene mixture and lead to homogeneous gene pools among populations [48]. It is also known that two major spruce species in Hokkaido, *P. jezoensis* and *P. glehnii*, demonstrate homogeneous gene pools in Hokkaido [49,50]. In this regard, anemogamous conifer species can be said to generally show homogeneous gene pools among local populations in Hokkaido.

However, there is evidence of genetic differentiation between the southwestern populations and the rest of the populations. The results from PCA revealed that the southern peripheral populations were differentiated from the other central populations (Figure 4). Geographic isolation of some

populations was reflected in the PCA. P20 is located in a volcanic region so that the forest soil is composed of volcanic ash or pumice as the base materials. Due to recent volcanic activity, the continuity of the species distribution might also be interrupted. P21 is located in a deep ravine at the tip of a peninsula and P19 in a coastal area, with both populations being isolated from other *A. sachalinensis* populations. Therefore, P19, P20, and P21 were plotted at the periphery of the PCA plane due to their geographic isolation (Figure 4).

The phylogenetic tree indicated that the six southernmost populations, P20 to P25, were differentiated from the other populations (Figure 5). These results indicated that the populations of the southernmost distribution were highly differentiated from the other populations. This finding is consistent with the previous studies based on morphological traits [34], allozyme variations [21], and organelle DNA haplotypes [51].

The high genetic divergence of southernmost populations may be also explained by the fossil pollen record in southwestern Hokkaido [52], which revealed that *Abies*, supposedly *A. sachalinensis*, that had dominated sub-boreal forests was completely replaced by cool temperate forest at the end of the last glacial period (10,000 yrs. BP), indicating that the southwestern populations became smaller and more isolated from each other at that time. Pairwise differentiation by $G'_{ST}$ indicated significantly high values in the southwestern populations (Table S3). These relic populations might have lost gene exchange with the distributional center. This would cause a genetic drift that affected the genetic structure of the small, isolated populations on the distribution periphery. In addition, southwestern Hokkaido is made up of a long, narrow peninsula, which can be a topographical barrier against the frequent exchange of individuals.

In conclusion, the central-marginal hypothesis that marginal populations show less genetic variation and higher differentiation than do central populations [4–6] was found to be relevant to natural populations of *A. sachalinensis* in Hokkaido, with the southwestern populations being highly differentiated from the other populations. In addition, the longitudinal genetic cline revealed by Nagasaka et al. [21] was supported by the 19 EST-SSR markers in this study. This cline may be related to adaptation to the environmental gradient in Hokkaido.

## 5. Conclusions

We analyzed the genetic diversity of 25 natural populations of a major sub-boreal conifer, *Abies sachalinensis*, including the species distribution range from its center to its southern margin. Nineteen EST-SSR loci were applied and revealed that the genetic diversity parameters were higher among the eastern populations and lower among the southwestern ones. This result supported the central-marginal hypothesis that the distribution center possesses higher gene diversity because the eastern populations are located in the core of species distribution. Phylogenetic analysis revealed that the marginal populations at the southern range limit showed further genetic distances from the central populations. The eastern to southwestern gradient of genetic diversity indicated a relationship to the species' adaptation to certain environmental factors.

**Supplementary Materials:** The following are available online at http://www.mdpi.com/1999-4907/11/2/233/s1, Figure S1: Genetic diversity parameters for the 25 populations. Bars indicate the 95% confidence intervals, Figure S2: Relationships among number of alleles and longitude east. Lines indicate the least squares, Figure S3: Relationships among effective number of alleles and longitude east. Lines indicate the least squares, Figure S4: Relationships among $H_E$ and longitude east. Lines indicate the least squares, Figure S5: Relationships among $G_{IS}$ and longitude east. Lines indicate the least squares, Figure S6: STRUCTURE results from K = 2 to 4. Populations are arranged from SW (left) to NE (right), Table S1: Climatic conditions for the 25 populations of *A. sachalinensis* used in this study, Table S2: Gene diversity among 19 EST-SSR loci used in this study, Table S3: Pairwise differentiation matrices by $G'_{ST}$ (lower triangle) and *p*-values (upper triangle) between populations, Table S4: Correlation analysis between longitude and selected environmental factors for the 25 populations of *A. sachalinensis*, Table S5: Regression coefficients between environmental factors and the number of alleles for each locus.

**Author Contributions:** Conceptualization, G.S., K.K., and I.W.; methodology, K.K., U.K., and U.S.; formal analysis, K.K.; investigation, K.K., U.K., I.W., T.I., and G.S.; writing—original draft preparation, K.K.; writing—review

and editing, G.S., K.K., U.K., I.W., T.I., and U.S. All authors have read and agree to the published version of the manuscript.

**Funding:** This research was funded by a Grant-in-Aid for Scientific Research from the Japan Society for the Promotion of Science, grant number 16H02554 and 16H06279 (PAGS).

**Acknowledgments:** We thank T. Kawahara and A. Takazawa for their technical support.

**Conflicts of Interest:** The authors declare no conflict of interest.

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
