# Peer review of "Geographical Gradients of Genetic Diversity and Differentiation among the Southernmost Marginal Populations of Abies sachalinensis Revealed by EST-SSR Polymorphism"

_forests, doi:10.3390/f11020233_

Round 1
Reviewer 1 Report
The authors used microsatellite genotyping at 19 EST-SSR loci to evaluate the genetic diversity and differentiation within and among 25 natural populations of Abies sachalinensis, sampled across the island of Hokkaido. Genetic diversity proved to be high in all populations but the authors also found a slightly lower diversity in the south-westernmost populations as compared with the remainder of the sampling. Since the latter pops were sampled along the margins of the distribution range of the species, the authors conclude that their results lend support to the “central-marginal” hypothesis” of gene diversity for Abies sachalinensis. The paper is reasonably well written and, as far as I can judge, the English is mostly fine. The experimental part of the study is correctly designed and technically sound. The data are mainly presented in an appropriate form, but some are missing (see below). In my view the primary data do not convincingly justify the main conclusion of the authors concerning the existence of a geographical gradient (see below). My recommendation is that the manuscript should become acceptable for publication provided the authors consider the following suggestions for improvement.
My main point is that the authors should be much more cautious with their interpretations. Thus, I see only very slight differences in genetic diversity among populations, no matter which parameter is considered (Table 1). Although Figs. 2, 3 and S2-S4 show that there is a certain tendency of decreasing diversity from NE to SW, absolute values are within a narrow range and only very few of the observed differences among populations are significant (see Fig. S1). The authors themselves admit that some individual EST-SSR loci in fact suggest an opposite tendency for the effective number of alleles, Fig. S3, and for He, Fig. S4). So I am afraid the authors somewhat overinterpret their observations.
The same is true for the extent of genetic differentiation. In l. 217 ff., the authors make the statement that the southernmost populations show a strong genetic differentiation from the remaining populations. This was apparently only concluded from their outlier positions in the PCoA and the NeighborNet. The differentiation of southern pops in these graphs is however not so clear-cut as the authors claim. Thus, P1 and P2 are geographically very close to P20 and P21 but do not show an outlier position in the diagrams, whereas P24 is geographically far SW, but genetically not so isolated. Also with regard to the G´ST values among pops summarized in l. 124-125, I would not consider the results as indicative for a “strong genetic differentiation” between SW pops and remaining pops. I also cannot find a list of G´ST values anywhere in the manuscript. I suggest that the authors provide a matrix of pairwise G´ST and/or FST values between populations to support their conclusion of a stronger genetic isolation of SW populations. In this respect, it would also be interesting to see the results of an AMOVA, based on two or better three hierarchy levels: within pops, among pops within regions (i.e. NE and SW), among regions.
Overall the main result of the study seems to be that genetic diversity within populations is high and genetic differentiation between populations is low, indicative of high gene flow among populations (presumably mainly mediated by pollen). This is also illustrated by the inability of STRUCTURE to reveal genetic clusters and is the expected result for wind-pollinated conifers. There might be a slight gradient of decreasing genetic diversity from NE to SW, but this is not well-supported by the data and therefore not a primary result
Some minor points, as they appear in the text:
99-107: Be consistent: QIAGEN or Qiagen? 104 ff. Give some more details for marker characteristics by including them in Table S1 (see below) 105: Skip “(Accession number will be provided soon)” This information is provided twice here. 112: Include the abbreviations (in parentheses) for number of alleles, effective number of alleles, and Nei’s genetic distance in the text 115: Explain in more detail how a contour map is laid out. Are there any references for applications of this approach in population genetics except for the “R core team 2016”? 119 Table 1. Include in Table legend “…and genetic diversity parameters…” I would prefer to also see the absolute (total) numbers of alleles in each population (column Na) rather than the average values across loci 120 Table 1: Use superscript for numbers 3 and 4 124-125 The statement “The G’ST ranged from -0.105 to 0.110, and the overall value was 0.016 (Table 1)” is not correct since there are no G´ST values presented in Table 1. Where can they be found? 182 Write out in full: Fagus crenata 183-185: “This might be related to population size in that the small and isolated populations in distribution margins receive a relatively high amount of gene flow from outside of the population, resulting in outbreeding”. I am not so convinced by this argument. Isn´t it usually the other way round, i.e., small and isolated populations are often characterized by genetic drift and increased inbreeding because of limited interpopulational gene flow (this is why they are called “isolated”). Do we actually know anything about population size of marginal vs. central pops of A. sachalinensis in Hokkaido? That is, are SW populations really smaller (or less well connected) than the NE ones? Overall, I wouldn´t give too much attention to the meaning of the GIS values found here, given that most of them are anyway negligibly low. 219 Be more specific here, perhaps reword: ”…that the six southernmost populations P20-P25 were differentiated from all other populations…”. This differentiation is however not so clear-cut (see above).In Table S1 some important locus characteristics should be added, including the microsatellite motifs, the primer sequences and the GenBank accession numbers, especially since the Egm loci are apparently new and have not been published before).
Author Response
Dear Reviewer 1
We would like to thank for constructive comments on our manuscript. We have responded and revised the manuscript according to these comments and are resubmitting the amended version. Point-by-point responses were listed in the attachment file.
Sincerely,
Susumu Goto

Reviewer 2 Report
The topic of the manuscript is interesting, Authors studied the genetic diversity marginal populations of Abies sachalinensis. The spatial distribution of genetic variation could provide imprtant informations for forest tree conservation and management.
They detected the longitudinal gradients of genetic diversity parameters (alleles number, heterozygosity, inbreeding coefficient) in A. sachalinensis Hokkaido population. They ipotized that these could be attributable to climatic adverse conditions. Polymorphism study was performed via EST-SSR that is a valid and reproducible tecnique.
Authors should better explain why thay saied that genetic diversity could be related to climatic condition. Thay do not report any more information about in the manuscript. The only reference is in the introduction: Hatakeyama [16] reported that regional differences in morphological traits
were closely related to the snow-related climatic conditions based on a common garden experiment (line 74-75). The influence of climatic conditions on agronomical and morphological traits is well known, indeed the reported reference is of 1981 year. Some evidence of climatic conditions-genetic polymorphisms should be reported and discussed.
However manuscript need of an english language editing and authors should report correctly the scientific name of the mentionated species. Some information on the ecological importance of the chosen specie could be important to better inquadring the research scope.
Discussion section should be improved. Some new references are required for a correct results presentation and discussion.
Author Response
Dear Reviewer 2
We would like to thank for constructive comments on our manuscript. We have responded and revised the manuscript according to these comments and are resubmitting the amended version. Point-by-point responses are listed in the attachment file. We hope you find our revised manuscript meets the standard for publication in Forests.
Sincerely,
Susumu Goto

Round 2
Reviewer 1 Report
The authors carefully addressed all points that I raised in my review of the first version of the manuscript. The paper has now become much more clear, straightforward and comprehensible. The authors also my suggestions to include additional data and calculations, like the Tables with the pairwise genetic distance values and the results of the AMOVA. In my view, the revised manuscript can be published almost as it stands. I have only a few editorial suggestions:
33. The authors accepted my point of criticism that the genetic differentiation is not so high, and changed the wording in the text accordingly (new in L.334-35: “However, there is evidence of genetic differentiation between the southwestern populations and the rest of the populations.”). A similar statement should also be made in the Abstract (e.g. replace “highly” by “genetically”). 92-93 What happened with some of the Figures? Look somehow duplicated 241 …between solar radiation… 310 Change to either “heterozygosity of individuals” or “heterozyosity of an individual” 312. Populus tremuloides 473. There are some erroneous capitals in the names “Martínez” and “Penuelas” 474, 482, 487-88. italicize scientific names of species and generaTable S2, correct citation is “Postolache et al. 2014”. I would actually suggest to include the primer sequences and microsatellite motifs also for the previously published AAT loci in the Table, so that the reader can easily compare the locus characteristics (e.g. motif type and length) of both marker sources. Another column coined “Origin” could then be added, with the entries “Postolache et al. 2014” and “this study”, respectively
Author Response
Dear Reviewer 1
We would like to thank you for their constructive comments on our manuscript. We have responded and revised the manuscript according to the comments and are resubmitting the amended version. We hope you find our revised manuscript meets the standard for publication in Forests.
Sincerely,
Susumu Goto
Reply to Comments and Suggestions
Comment: 33. The authors accepted my point of criticism that the genetic differentiation is not so high, and changed the wording in the text accordingly (new in L.334-35: “However, there is evidence of genetic differentiation between the southwestern populations and the rest of the populations.”). A similar statement should also be made in the Abstract (e.g. replace “highly” by “genetically”).
Response: We have incorporated the recommended change to the text (L32).
Comment: 92-93 What happened with some of the Figures? Look somehow duplicated
Response: We have asked the Assistant Editor about the appearance of Figure 1. We understood why this problem is occurred. According to the tracking system, it looked duplicated. She has confirmed that the Figure 1 appears properly formatted on the file.
Comment: 241 …between solar radiation…
Response: We have incorporated the recommended change to the text (L183).
Comment: 310 Change to either “heterozygosity of individuals” or “heterozyosity of an individual”
Response: We have incorporated the recommended change to the text (L244).
Comment: 312. Populus tremuloides
Response: We have corrected the error (L246).
Comment: 473. There are some erroneous capitals in the names “Martínez” and “Penuelas”
Response: We have corrected the errors (L405).
Comment: 474, 482, 487-88. italicize scientific names of species and genera
Response: We have corrected the errors (L414, L419-420).
Comment: Table S2, correct citation is “Postolache et al. 2014”. I would actually suggest to include the primer sequences and microsatellite motifs also for the previously published AAT loci in the Table, so that the reader can easily compare the locus characteristics (e.g. motif type and length) of both marker sources. Another column coined “Origin” could then be added, with the entries “Postolache et al. 2014” and “this study”, respectively
Response: We have incorporated the recommended changes in the Table S2.